# Potential Benefits of Lycopene Consumption: Rationale for Using It as an Adjuvant Treatment for Malaria Patients and in Several Diseases

**DOI:** 10.3390/nu14245303

**Published:** 2022-12-14

**Authors:** Everton Luiz Pompeu Varela, Antônio Rafael Quadros Gomes, Aline da Silva Barbosa dos Santos, Eliete Pereira de Carvalho, Valdicley Vieira Vale, Sandro Percário

**Affiliations:** 1Oxidative Stress Research Laboratory, Institute of Biological Sciences, Federal University of Pará, Belém 66075-110, Brazil; 2Post-Graduate Program in Biodiversity and Biotechnology of the BIONORTE Network, Federal University of Pará, Belém 66075-110, Brazil; 3Post-Graduate Program in Pharmaceutical Innovation, Federal University of Pará, Belém 66075-110, Brazil

**Keywords:** lycopene, malaria, oxidative stress, carotenoids, supplementation, antioxidants, adjuvant treatment

## Abstract

Malaria is a disease that affects thousands of people around the world every year. Its pathogenesis is associated with the production of reactive oxygen and nitrogen species (RONS) and lower levels of micronutrients and antioxidants. Patients under drug treatment have high levels of oxidative stress biomarkers in the body tissues, which limits the use of these drugs. Therefore, several studies have suggested that RONS inhibition may represent an adjuvant therapeutic strategy in the treatment of these patients by increasing the antioxidant capacity of the host. In this sense, supplementation with antioxidant compounds such as zinc, selenium, and vitamins A, C, and E has been suggested as part of the treatment. Among dietary antioxidants, lycopene is the most powerful antioxidant among the main carotenoids. This review aimed to describe the main mechanisms inducing oxidative stress during malaria, highlighting the production of RONS as a defense mechanism against the infection induced by the ischemia-reperfusion syndrome, the metabolism of the parasite, and the metabolism of antimalarial drugs. Furthermore, the effects of lycopene on several diseases in which oxidative stress is implicated as a cause are outlined, providing information about its mechanism of action, and providing an evidence-based justification for its supplementation in malaria.

## 1. Introduction

Malaria is currently endemic in 85 countries and is found on most continents, but it is mostly confined to tropical and subtropical regions. It is noteworthy that repeated *Plasmodium* infection does not result in complete immunity, so populations in endemic regions are continuously susceptible to infection, transmission, morbidity, and mortality. Moreover, the lack of effective prevention strategies, including medications and/or vaccines, contributes significantly to this scenario. In 2020, there were 241 million cases, and 627,000 people died worldwide from the disease [1]. Almost all malaria-related deaths result from *Plasmodium falciparum* infection.

The pathophysiological mechanisms involved in the disease are complex and multifactorial. Inflammatory molecules are greatly involved and related to several cell signaling pathways. Indeed, after *Plasmodium* infection, an inflammatory reaction may be observed, with a predominance of neutrophils, lymphocytes, and monocytes, which are attracted by the presence of the parasite in the body [2,3,4]. Furthermore, leukocytes induce the expression of proinflammatory cytokines, including interleukin (IL)-1β, IL-2, IL-6, IL-17, interferon-γ (IFN-γ), and tumor necrosis factor-α (TNF-α), that play an important role in the protection against malaria and elimination of parasites, by inducing monocyte phagocytosis, favoring the elimination of parasitized erythrocytes and limiting the progression of uncomplicated malaria to malaria with serious complications [5,6,7].

Additionally, in the recovery phase, regulatory cytokines, including IL-4, IL-10, chemokines, including IL-8, macrophage inflammatory protein (MIP)-1α, MIP-1β, macrophage colony-stimulating factor (M-CSF), and granulocyte-macrophage colony-stimulating factor (GM-CSF) [8,9,10,11] and transforming growth factor-β, neutralize the pro-inflammatory response by inhibiting the production of T helper 1 cytokines, contributing to the elimination of the parasite and reducing the risk of serious clinical complications [12,13,14].

However, a disturbance in the balance of pro- and anti-inflammatory cytokines and the underlying inflammatory process has been implicated in the pathogenesis of cerebral malaria and is associated with disease severity and death [11,15,16]. Such disturbance may be promoted by oxidative stress, which is known to intensify inflammation through tissue destruction and the release of danger signals by necrotic cells [17,18]. According to Ty et al. [19], reactive oxygen and nitrogen species (RONS) play an important role in triggering inflammation in malaria since these are produced in excess during infection and are potent inducers of inflammatory cytokines, suggesting the important role of oxidative stress in the pathophysiology of the disease [19,20,21].

Given the tropism of *Plasmodium* species for tissues such as blood [22], important systemic effects, including the induction of cytokines and RONS, which are closely associated with anemia, paroxysms, cerebral malaria, among other symptoms of systemic infection, are marked during the disease [20,23,24,25].

The oxidative changes occurring during infection that led to oxidative stress are a result of several different mechanisms, including the degradation of hemoglobin by the malaria parasite, producing redox-active by-products, such as free heme and hydrogen peroxide (H_2_O_2_) [26]. These radicals stimulate a series of oxidative reactions, leading to a decrease in the antioxidant defense system, through the consumption of micronutrients, including vitamin A, zinc, ascorbic acid (vitamin C), α-tocopherol (vitamin E), and carotenoids, among others [27]. In fact, in malaria-endemic areas, *P. falciparum*-infected individuals present lower plasma concentrations of various micronutrients compared to healthy individuals [28].

On the other hand, these micronutrients play essential roles in the antioxidant system and are implicated in resistance to malaria infection [29]. In this sense, it has been shown that vitamin A, zinc, and selenium can interfere with the progression of oxidative reactions during malaria in mice infected with *P. berghei* [30]. Additionally, studies have suggested that the periodic supplementation of vitamin A and zinc can reduce the incidence of febrile episodes and parasitemia, being an effective and low-cost strategy to decrease *P. falciparum* morbidity in preschool children [31,32].

Other studies also support the hypothesis that the use of carotenoids by the host increases during malaria, suggesting that the nutritional status is an important modulating factor in the disease [28,33]. In this regard, it has been suggested that increased plasma lycopene concentration is associated with faster resolution of parasitemia in children infected with *P. falciparum*, being effective in maintaining the oxidative balance [34].

Considering the important involvement of oxidative stress mechanisms in malaria and, therefore, the potential of antioxidant nutrients in preventing it, in the present revision, we intend to demonstrate the beneficial effects of lycopene supplementation in malaria patients and, consequently, in several other diseases mediated by oxidative stress.

## 2. Oxidative Stress

Oxidative stress occurs when RONS overwhelm cellular defenses, causing damage to proteins, membranes, and deoxyribonucleic acid (DNA) [35]. It is the result of a disturbance in the balance between RONS and antioxidants in favor of RONS [36]. Under physiological conditions, endogenous RONS are generated by enzymatic systems, including nicotinamide adenine dinucleotide phosphate oxidase (NADPH oxidase) and nitric oxide synthase (NOS), as a by-product of mitochondrial electron transport chain reactions (Figure 1) or by metal-catalyzed oxidation [37,38].

In this regard, the free radical superoxide (O_2_^•−^), resulting from the monoelectronic reduction of oxygen, is considered the main precursor of other RONS since, after its formation, it can react with other molecules giving rise to other free radicals, such as hydroxyl (OH^•^), alkoxyl (RO^-^), and peroxyl (ROO^-^), in addition to other molecules that do not meet the definition of free radicals, but take part of oxidative reactions in a meaningful way, such as H_2_O_2_. Nitric oxide (NO) is among the molecules that can react with O_2_^•−^, and the reaction between them generates the free radical peroxynitrite (ONOO^-^). Additionally, O_2_^•−^ can be unmuted to form H_2_O_2_, and it can be broken down through Fenton or Haber-Weiss reactions, leading to the generation of OH^•^ [39,40].

These RONS-generating chain reactions are initially controlled by antioxidant defense systems that act quickly, neutralizing any molecule that can potentially develop into a RONS or any free radical with the ability to induce the production of other pro-oxidants [41]. Three enzymes are critical in this process, including superoxide dismutase (SOD), catalase (CAT), and glutathione peroxidase (GSH-Px). These enzymes, respectively, unmute O_2_^•−^ and break down H_2_O_2_ or hydroperoxides (ROOH) into harmless molecules such as H_2_O, alcohol, and oxygen (O_2_) [42]. The class of endogenous antioxidants also includes glutathione reductase, and reduced glutathione (GSH), in addition to small molecules such as coenzyme Q and uric acid (UA), among others [43]. Since they can be synthesized by the body in response to oxidative aggression, we nominate endogenous antioxidants as *mobilizable antioxidant molecules*.

However, in diseases in which oxidative stress is a pathogenic mediator, including cancer and malaria, mobilizable antioxidants are not sufficient to maintain cell homeostasis due to the decreased synthesis of antioxidant enzymes and increased use of these antioxidants, among other factors [44,45,46].

In these cases, supplementation with dietary antioxidants is essential to maintain optimal cell function. Vitamins, including vitamins E and C, phenolic substances, such as flavonoids, resveratrol, and carotenoids, including β-carotene and lycopene, and drugs, such as *N*-acetylcysteine (NAC), among others, belong to this category [47,48]. Dietary antioxidants neutralize or eliminate RONS by binding or donating electrons to pro-oxidants, and in the process, they become free radicals but with less harmful effects. These “new radicals” are more easily neutralized and rendered completely harmless by other antioxidants in this group [49]. Thus, this class of antioxidant molecules can also be referred to as *consumable antioxidants*, as they are consumed in the face of oxidative aggression. Thus, consumable and mobilizable antioxidants act synergistically to fight the excessive increase in RONS, which can be a primary cause or a secondary complication of various diseases [50,51], as in malaria [52,53].

## 3. Oxidative Stress in Malaria

In malaria, oxidative stress is caused by four main mechanisms: a host defense against *Plasmodium* infection; ischemia-reperfusion syndrome; direct production of oxidative species by the parasite; and the metabolism of antimalarial drugs [54].

### 3.1. Oxidative Stress as a Host Defense Mechanism

RONS are essential for several physiological functions of the body, including cell survival, growth, proliferation, and differentiation, as well as the immune response [55,56].

As for the immune response, RONS are important for phagocytes, including neutrophils and monocytes/macrophages, which are highly activated during malaria, helping these cells phagocytize and destruct parasites [57,58].

In this sense, the body’s defense system responds to infection by primarily recruiting neutrophils [59]. When neutrophils engulf the parasites, they induce a respiratory burst (Figure 2), in which O_2_ enzymatically reacts with NADPH oxidase present in the plasma and the phagosomal membrane of neutrophils, forming O_2_^•−^ [4]. O_2_^•−^ and its derivatives H_2_O_2_ and OH^•^, when released by activated neutrophils in the phagosome, are essential to kill ingested pathogens [60].

In addition, activated neutrophils produce cytokines, such as GM-CSF and M-CSF, and chemokines, including MIP-1α and MIP-1β, which attract these cells and are essential for monocyte mobilization [61]. These leukocytes engulf and kill the parasites through the oxidative action of O_2_^•−^, which is generated in the same way as in neutrophils, as well as by the action of NO, which is produced by the macrophage from the reaction of NOS with L-arginine [62,63]. Additionally, the NO and O_2_^•−^ generated react to form other RONS, such as ONOO^-^, intensifying the cytotoxicity directed against the parasites [64].

Furthermore, neutrophil and macrophage myeloperoxidase is activated and uses H_2_O_2_ as a substrate to produce hypochlorous acid, a highly bactericidal compound [65,66]. On the other hand, phagocytosis and the consequent action of RONS, including O_2_^•−^ and NO, as well as other toxic products, can exacerbate the condition due to rupture of the parasitized erythrocytes, during which normal uninfected erythrocytes can also be destroyed, stimulating cytoadherence and, consequently, potentially blocking blood flow, causing ischemia and anemia [67,68].

### 3.2. Oxidative Stress Due to Ischemia-Reperfusion Syndrome

In individuals with malaria, severe anemia induces microvascular dysfunction, leading to recurrent episodes of initial restriction of blood supply to organs, which can lead to ischemia and nutrient and oxygen deprivation, followed by subsequent restoration of concurrent perfusion and reoxygenation [54,69]. This process is called ischemia-reperfusion syndrome and can occur in malaria due to the sequestration of parasitized erythrocytes, as a result of the destruction of erythrocytes caused by the parasites and RONS during the paroxysm of malaria, and due to cytoadherence of erythrocytes to blood vessels [70].

Furthermore, this syndrome can trigger anaerobic metabolism, the production of lactic acid, and the consequent depletion of adenosine triphosphate (ATP). As ATP availability is reduced, ATP-dependent ion channels begin to fail. At the same time, calcium overload and excessive RONS production open the mitochondrial permeability transition pore, further reducing ATP levels [71,72]. During the ischemic process, the degradation of ATP causes the accumulation of xanthine oxidase (XO) and hypoxanthine due to the lack of oxygen. When the blood supply is resumed, XO acts on hypoxanthine resulting in the production of O_2_^•−^ (Figure 3), which can later be converted into OH^•^ in the presence of transition metals and, consequently, trigger oxidative stress [73,74].

During this process, UA is also formed, which is a weak organic acid present mainly as monosodium urate at physiological pH [75]. UA can be found in the host organism during malarial infection and can act by eliminating RONS and chelating transition metal ions or even by reducing NOS expression, impairing NO release [76,77]. Previous studies have shown that plasma UA levels in *P. falciparum*-infected children increase during acute episodes and with disease severity, suggesting that UA is an important mediator in the pathophysiology of malaria [78,79].

In the ischemia-reperfusion syndrome, RONS can be produced during ischemia but is massively increased during reperfusion, amplifying and propagating oxidative damage and destroying the integrity of proteins, membranes, and microvascular endothelium [80].

### 3.3. Oxidative Stress Due to the Metabolism of the Parasite

Another important oxidative mechanism in malaria is mainly triggered by the metabolism of the parasite, as well as by the potentially oxidative by-products generated and released from red blood cells destroyed by the action of the parasite [81]. Inside the erythrocyte, the parasite digests hemoglobin in its acidic digestive vacuole, forming essential amino acids for parasite development and proliferation [82]. However, in this process, ferroprotoporphyrin IX or heme complex (FPIX) is released, which is toxic to the parasite. On the other hand, this complex can still be detoxified within the parasite by polymerization [83].

Although the parasite manages to polymerize FPIX, resulting in a nontoxic derivative, hemozoin, also known as a malarial pigment, a significant amount escapes polymerization [84]. Thus, the ferrous iron (Fe^2+^) from FPIX is oxidized to the ferric state (Fe^3+^), with the consequent production of superoxide, which dismutates to H_2_O_2_ (Figure 4). This oxidative reaction chain leads to the production of OH^•^ from reactions involving H_2_O_2_ and Fe^3+^, such as the Fenton and Haber–Weiss reactions [85].

These free radicals can cause damage to the parasite’s digestive vacuole membrane, eventually killing it [81]. However, the rapid development and proliferation of the parasite, associated with the RONS generated and released inside the erythrocytes, cause structural damage to the erythrocytes [86]. This results in increased membrane permeability for ions, increased cell volume, oxidation of sulfhydryl groups, and reduced deformability, contributing to the loss of erythrocyte function and cell lysis [67,87].

Consequently, all intra-erythrocyte content, including RONS, will be released to the extracellular environment, resulting in damage to several biomolecules, such as lipids, proteins, and DNA, as well as enzyme inactivation, apoptosis induction, modification of surface adhesion molecule expression of leukocytes and endothelial cells, and alteration in the bioavailability of NO, compromising homeostasis and, ultimately, its survival [88,89]. These changes expose the host organism to a highly oxidative environment (Figure 5), implying the development of systemic complications such as reduced blood flow and severe anemia and also facilitating the entry of parasites into tissues such as the lung and brain, which can lead to organ failure [83,90,91,92].

### 3.4. Oxidative Stress as a Consequence of the Metabolization of Antimalarial Drugs

The drug treatment of malaria is specially designed to interrupt parasite proliferation, responsible for the pathogenesis and clinical manifestations of the infection, to destroy the latent forms of the parasite (hypnozoites) to prevent late relapses, and to prevent the transmission of the parasite, through the use of drugs that prevent the development of sexual forms of the parasites [93,94].

In this context, one of the main targets of antimalarial drugs is the intracellular pathway of heme metabolism, which is implicated in the production of RONS and the consequent death of the parasite [95,96]. Therefore, chloroquine, a quinoline blood schizonticidal drug used to treat severe and uncomplicated cases of malaria, can act by preventing FPIX polymerization, causing the accumulation of FPIX in the parasite’s digestive vacuole and consequent lethal oxidative stress in the parasite [97,98,99]. However, there are increasing reports of *P. falciparum* resistance to quinoline antimalarials, highlighting the importance of the *P. falciparum* chloroquine resistance transporter, a member of the drug/metabolite transporter superfamily located in the parasite’s digestive vacuole, as the main responsible for chloroquine resistance [100,101,102].

Other studies indicate that, in addition to showing chemical similarity with chloroquine and a similar mechanism of action, other quinolines, such as quinine, amodiaquine, lumefantrine, and mefloquine are effective against many strains of parasites resistant to chloroquine [103,104,105]. In addition, some of these drugs are widely used in combination therapies with artemisinin derivatives, including artemether plus lumefantrine and artesunate plus amodiaquine, and provide synergistic antimalarial activity along with preventing the development of antimalarial drug resistance [106,107,108].

The site of action of artemisinin and its derivatives dihydroartemisinin, artemether, arteether, and artesunate is believed to be the parasite’s digestive vacuole, where these drugs can interfere with the FPIX complex, giving rise to RONS, leading to damage to nearby proteins, and still interacting with the mitochondrial electron transport chain of the parasite, enhancing RONS production, impairing mitochondrial functions, and killing the parasite [109,110,111].

Artemisinins act quickly and are very potent against blood-stage parasites. They are active against the sex stages of the parasite, which is important for blocking transmission [112,113]. However, due to their short half-life, these drugs are used in conjunction with other long-acting drugs that remain in the body for longer to fight potential remaining parasites [114,115].

Accordingly, studies show that primaquine increases the effect of combination therapy with artemisinin derivatives in eliminating malaria and reduces the risk of artemisinin-resistant infections [116].

Furthermore, only primaquine is recognized for completely eliminating *P. vivax* and *P. ovale* that form hypnozoites—the latent form of the parasite that remains in the liver and is responsible for disease relapse in individuals infected by these parasites—refractory to most drugs and for providing a radical cure [93,117]. Primaquine, an 8-aminoquinoline, can act directly on erythrocytes leading to massive production of RONS and consequent lipid peroxidation of the cytoskeleton and membrane, as well as hemolysis [118]. However, the use of primaquine in individuals with glucose-6-phosphate dehydrogenase deficiency can result in clinical manifestations of hemolysis, such as severe anemia, fatigue, jaundice, and acute renal failure, thereby limiting its use [119,120].

In this scenario, as a product of the normal host’s metabolism or from the metabolism of the parasite, or as an effect of pharmacological treatment, intensely produced RONS cause damage to lipids, proteins, and DNA, leading to oxidative stress that impairs the normal functioning of the infected organism [98]. Therefore, the search for adjuvant therapies that can improve the clinical outcomes of malaria continues because, despite their benefits, treatments eventually cause oxidative damage, which limits their use [121].

### 3.5. Nitric Oxide in Malaria

Scientific evidence demonstrates that a specific RONS is particularly involved in the pathophysiology of this disease: NO [122,123]. It has been suggested that the low bioavailability of NO promotes oxidative stress in tissues such as the brain and lungs [124]. On the other hand, it has been shown that NO at high concentrations can kill *Plasmodium* [125,126]. NO is an important mediator of biological processes such as vascular homeostasis, neurotransmission, immunity, and inflammation [127,128,129]. Furthermore, it is a free radical produced by three different nitric oxide synthase enzymes, neuronal NOS (nNOS or NOS1), endothelial NOS (eNOS or NOS3), which are constitutively expressed, and the inducible NOS (iNOS or NOS2), which is induced by inflammatory stimuli [130,131,132,133]. NO is very reactive and has a very short half-life. For this reason, nitrite and nitrate measurements, which are the final metabolites of NO, have been used to measure the concentration of NO indirectly [134,135].

Experimental evidence indicates that NO plays an important role in the defense against plasmodia in vitro and in vivo [136,137]. In this context, studies have shown that circulating levels of nitrite and nitrate were higher in anopheline mosquitoes—a natural vector of malaria in humans—infected with *Plasmodium* and that increased NO concentrations at the beginning of the sporozoite stage induced the formation of toxic metabolites, limiting parasite development [138].

In children and adults with malaria, elevated plasma levels of nitrites and nitrates have been associated with more rapid parasite clearance [139]. Indeed, previous studies have shown that children infected with *P. falciparum* had elevated levels of NO and iNOS activity, suggesting the protective role of NO in children with malaria [140]. Protection against severe malaria in this population of children appears to be associated, at least in part, with a polymorphism in the iNOS gene, which produces high levels of NO during an inflammatory event [141]. These studies suggest that NO production during malaria depends on the severity of the disease and the degree of patient immunity [142].

In an animal model of experimental cerebral malaria (ECM), Serghides et al. [143] demonstrated that pretreatment with inhaled NO reduced the accumulation of parasitized erythrocytes in the brain, decreased endothelial cell expression, and preserved vascular integrity. From these results, the authors suggested that prophylaxis with NO inhalation can reduce systemic inflammation and endothelial activation during ECM. In a similar model, Ong et al. [144] showed that cerebrovascular dysfunction is characterized by vascular constriction, occlusion, and cell damage, resulting in impaired perfusion and reduced cerebral blood flow and oxygenation, and was associated with low NO bioavailability.

Given the critical importance of NO-derived and -non-derived oxidative stress in the underlying pathophysiological mechanisms of the disease, studies have shown that natural or synthetic exogenous antioxidants, including vitamin A, E, zinc, selenium, NAC, curcumin, *Agaricus sylvaticus* mushroom, and carotenoids, can benefit the treatment of malaria [145,146,147,148]. Several studies have indicated an association between the use of carotenoids and a decrease in oxidative changes, suggesting that the antioxidant properties of these compounds are an important factor against malaria-induced oxidative stress [149,150]. The recent interest in carotenoids has focused on the role of lycopene in human health [151,152].

## 4. Lycopene

Lycopene is a natural constituent synthesized by plants and microorganisms [153]. It is a red pigment found in some fruits and vegetables, such as guava, watermelon, papaya, pitanga *(Eugenia uniflora—Myrtaceae)*, tomatoes, and their derivatives [154,155,156,157] and can be extracted from these vegetables by chemical reactions using organic solvents, such as ethanol and ethyl acetate and/or using a supercritical fluid such as supercritical carbon dioxide, or by heat treatment at different temperatures ranging from 60 to 140 °C [158,159,160,161,162,163]. It is widely used as a supplement in functional foods, nutraceuticals, and pharmaceuticals, as well as an additive in cosmetics [164,165].

Lycopene is an intermediate product of the β-carotene biosynthetic pathway that does not have provitamin A activity, as it does not have the β-ionone ring in its structure, which is responsible for this characteristic [166]. This compound is a noncyclic, fat-soluble hydrocarbon that contains 11 conjugated double bonds and 2 unconjugated double bonds, thereby offering it greater reactivity. This polyene can also exist in all-*trans* and *cis*-lycopene isomeric forms (Figure 6). Conversion from all-*trans*- to *cis*-lycopene forms can occur by geometric isomerism induced by light, thermal energy, or chemical reactions [167,168].

### 4.1. Sources

Tomatoes and tomato products are the main source of lycopene and are considered an important source of carotenoids in the human diet. In raw or fresh tomatoes, lycopene occurs mainly as a *trans* isomer [169]. However, *cis* isomers are better absorbed by the human body than *trans* isomers. *Cis* isomers form during cooking, food processing, and storage, which do not affect the total lycopene content [170].

Studies have shown higher plasma lycopene concentrations after ingestion of processed tomatoes compared to raw tomatoes [171,172]. In fact, processing and homogenization induce the disruption of the food membrane, converting lycopene from the *trans* to *cis* form, increasing its solubility and, consequently, its availability [173]. In addition, the acidic pH of the stomach also appears to secondarily contribute to such conversion, as it can lead to the transformation of the *trans* to *cis* form [174]. Thus, lycopene can be rapidly and completely absorbed without energy expenditure in the intestinal wall after oral administration in animals and humans [175].

### 4.2. Absorption

After absorption, lycopene can be found in high concentrations in human body fluids and tissues, such as breast milk, prostate, testis, and skin [176]. Furthermore, it is the predominant carotenoid in human plasma, naturally present in a higher concentration than β-carotene and other dietary carotenoids, which may indicate its greater biological significance for the human defense system [177].

Studies suggest that lycopene is transported between cells to target organs by specific proteins or migrates aggregated to chylomicrons, with the isometric form of lycopene being decisive for this process [178]. This is because, after passing through the stomach, trans isomers can readily aggregate within the intestine and form crystals, greatly reducing their absorption by micelles, while the cis form allows lycopene to be more efficiently incorporated into mixed micelles [179]. The lycopene-loaded micelles are then absorbed into enterocytes, from where they are released in chylomicrons, which exit to the lymph, passing from there to the systemic circulation to the liver. The liver stores and secretes carotenoids as very low-density lipoproteins (VLDL), which are subsequently absorbed by various tissues, including adrenal, kidney, adipose, splenic, lung, and reproductive organ tissues, and are subsequently recovered as other low-density (LDL) and high-density (HDL) lipoproteins [178,180]. During absorption, lycopene taken up by the enterocyte can also be cleaved by β-carotene 9′,10′-oxygenase (BCO2) to form apo-10′-carotenoid metabolites, including lycopene apo-10′-lycopenols [181,182,183].

### 4.3. Metabolism

Lycopene can be metabolized through isomerization, followed by oxidation to produce epoxides, or undergo eccentric cleavage by BCO2 to form apolycopenols [181,182,184,185]. Additionally, lycopene cleavage products can be generated through autoxidation, via reaction with free radicals [186], by processes that simulate biological tissues [187], or even by chemical reactions that cause the interruption of the polyene chain, affecting the carbon-carbon double bond system, and by addition or cleavage, resulting in several isomers and apolycopenols [159,188].

Among the most interesting lycopene metabolites formed by oxidative degradation of the hydrocarbon chain are the apolycopenols [189]. Apolycopenols have already been detected in animal tissues, such as ferret lungs [182] and the liver of rats [190], after treatment with lycopene. In addition, several apolycopenols have been isolated from fruits, vegetables, and human plasma [191].

Increasing evidence suggests that many of lycopene’s biological actions may be mediated, at least in part, by its metabolites and/or oxidation products [192,193,194]. In this regard, lycopenols were shown to reduce the proliferation of cancer cells, induce apoptosis, regulate the cell cycle, induce the expression of nuclear transcription factors, and enhance cell-to-cell communication [189,195,196,197,198]. Furthermore, the study by Böhm et al. [199] showed that the cis isomers obtained from processed foods had an antioxidant potential twice as intense as β-carotene. Additionally, studies by Lian and Wang [200] showed that treatment of human bronchial epithelial cells (BEAS-2B) with apo-10’-lycopenoic acid (10 µM) increased GSH levels and suppressed RONS production and oxidative damage induced by H_2_O_2_ in vitro. In addition, it was reported that apo-10’-lycopenoic acid induced the expression of phase II antioxidant enzymes mediated by factor 2-related nuclear erythroid factor 2 (Nrf2), including heme oxygenase-1 (HO-1), NAD(P)H quinone oxidoreductase 1 (NQO1), glutathione S-transferases (GST), GR, and γ-glutamylcysteine synthetase (γ-GCS) [200].

Apolycopenol treatment also inhibited methemoglobin-induced lipid peroxidation in a chemical model of postprandial oxidative stress in the gastric compartment [201].

In another in vitro study, both apo-10’-lycopenoic acid and apo-14’-lycopenoic acid inhibited RONS production and oxidative DNA damage induced by H_2_O_2_ and cigarette smoke. This effect was accompanied by the inhibition of mitogen-activated protein kinase (MAPK) phosphorylation, the expression of heat shock proteins (hsp)70 and hsp90, and the inactivation of NF-κB, molecules that are activated in situations of oxidative stress and which have also been implicated in the modulation of various intracellular redox functions [202].

## 5. Antioxidant Effects of Lycopene

Among the carotenoids, lycopene is the most effective antioxidant against RONS and may contribute to preventing or reducing oxidative damage to cells and tissues in vivo and in vitro [203]. Evidence supports the role of lycopene as a potent antioxidant, capable of scavenging singlet oxygen (^1^O_2_) and other free radicals, such as ROO^-^, with a potential twice as high as β-carotene, and ten times as efficient as α -tocopherol, although lycopene circulates at much lower concentrations than vitamin E [204,205]. During the elimination of ^1^O_2_, energy is transferred from this radical to the lycopene molecule and, as it has an open chain with 11 conjugated double bonds in its structure, this favors stabilization of the unpaired electron of the radical by resonance [206,207]. Additionally, it was observed that lycopene effectively eliminates other RONS, such as OH^•^, O_2_^•−^, and ONOO^-^ [208].

Furthermore, the lipophilic characteristic of lycopene favors its interaction with the lipid bilayer of the cell membrane, thereby preventing the breakdown of fatty acids and the oxidation of lipids, proteins, and DNA [180]. In this sense, Suwannalert et al. [209], investigating serum levels of lycopene and malondialdehyde (MDA) in elderly susceptible to oxidative stress, demonstrated that lycopene levels were inversely related to MDA levels. Additionally, Yonar and Sakin [206] demonstrated that lycopene treatment prevented deltamethrin-induced oxidative stress by decreasing MDA levels in fish (*Cyprinus carpio*) and significantly increasing SOD, CAT, and GSH-Px activities and the level of GSH. Similar results were found by Kujawska et al. [210], who reported that treatment with tomato extract enriched with lycopene was able to suppress the oxidative stress induced by *N*-nitrosodiethylamine in rats and increase the enzymatic antioxidant activity in these animals.

### 5.1. Cardioprotective Effect of Lycopene

Oxidative stress produced by RONS is implicated in the development of several diseases, including atherosclerosis and several heart diseases [50,211], but studies suggest that lycopene supplementation or consumption of tomato and its derivatives can improve endothelial function and lead to reduced blood pressure [187]. In this sense, Mohamadin et al. [212] investigated the cardioprotective potential of lycopene against isoproterenol-induced oxidative stress and cardiac lysosomal damage in rats. According to the authors, lycopene supplementation (4 mg/kg/day) significantly improved lysosomal membrane damage, as well as changes in cardiac enzymes, including aspartate aminotransferase, creatine kinase isoenzyme MB, and troponin T, as well as oxidative stress markers such as MDA, GSH, GSH-Px, SOD, and CAT.

Previously, Bose and Agrawal [213] had already observed, in a clinical study with grade I hypertensive patients, that tomato supplementation for 60 days improved the levels of antioxidant capacity and reduced lipid peroxidation in these patients. Ferreira-Santos et al. [151] reported that a lycopene-supplemented diet prevented angiotensin II-induced hypertension with no effect in normotensive rats. The authors suggested that the infusion of angiotensin II caused a decrease in the activity of antioxidant enzymes, and the treatment with lycopene improved the antioxidant balance, increasing the activity of GSH-Px and SOD, reducing oxidative stress, and improving cardiovascular remodeling. These results confirm the antihypertensive potential of lycopene without the risk of causing hypotension in normotensive individuals.

### 5.2. Anti-Atherosclerotic Effect of Lycopene

Atherosclerosis is a chronic inflammatory disease characterized by the accumulation of lipids and inflammatory cells in the walls of medium and large-caliber arteries, which is the main cause of heart disease and mortality in Western societies. The pathogenesis of atherosclerosis involves the activation of inflammatory mediators, cytokines, and increased oxidative stress [214,215].

In evaluating the effect of lycopene in an animal model of atherosclerosis, Renju et al. [216] demonstrated that CAT, SOD, and GSH-Px activities and GSH levels were increased, while the levels of thiobarbituric acid reactive substances (TBARS), total cholesterol, triglyceride, low-density lipoprotein (LDL), very-low-density lipoprotein, and inflammatory mediators, including cyclooxygenase-2 (COX-2) and 15-lipoxygenase, decreased after treatment with lycopene isolated from the alga *Chlorella marina*. Additionally, Martín-Pozuelo et al. [217] showed that tomato consumption improved the expression of genes such as fatty acid-binding protein 2, which encodes enzymes involved in lipid metabolism, thus reducing the synthesis of fatty acids, triglycerides, and cholesterol, preventing their accumulation and modulating the progression of steatosis induced in rats. Moreover, according to Navarro-González et al. [218], lycopene competes with hydroxymethylglutaryl coenzyme A in the liver, thus preventing the formation of mevalonate, and consequently reducing cholesterol synthesis by reducing the activity of the enzyme 3-hydroxy-3-methylglutaryl-coenzyme A reductase. For this reason, the consumption of tomato juice and the accumulation of lycopene in the liver were able to improve plasma cholesterol levels in steatosis induced in animals [218].

In this sense, Kumar et al. [219] observed that treatment with lycopene induced an increase in high-density lipoprotein and reduced levels of total cholesterol, LDL, triglycerides, and TBARS in rats fed a high-cholesterol diet. Brito et al. [220] also demonstrated that lycopene extracted from guava (*Psidium guajava* L.) reduced MDA and triglyceride levels, as well as reduced plasma activity of myeloperoxidase and hepatic steatosis in an animal model of dyslipidemia. The results indicated that lycopene has hypolipidemic and anti-atherogenic potential.

### 5.3. Hepatoprotective Effect of Lycopene

Oxidative stress is believed to be an important contributor to the pathogenesis of liver diseases, ranging from simple steatosis to its more severe form or even the genesis of hepatocellular carcinoma [221]. When investigating the role of lycopene in an animal model of hepatotoxicity, studies demonstrated that lycopene improved biochemical indices, both in the blood and in the liver of animals. Furthermore, lycopene restored the antioxidant capacity and increased the levels of GSH, GSH-Px, glutathione S-transferase (GST), CAT, and SOD, which, together with lycopene, could limit the production of oxidants [222,223,224]. Similar results were observed in the study by Abdel-Daim et al. [225], where zinc oxide poisoning in fish caused severe lipid peroxidation with a significant increase in the level of MDA in the liver, kidney, and gill tissues, and treatment with lycopene significantly reduced the production of this oxidative stress biomarker

Recently, Ni et al. [152] demonstrated that lycopene inhibited and reversed lipotoxicity-induced insulin resistance, preventing nonalcoholic steatohepatitis in mice, attenuating hepatic lipid accumulation, and increasing lipolysis. The beneficial effects of lycopene were attributed in part to decreased hepatic recruitment of T cells and macrophages, and to a reduction in macrophage M1/Kupffer cells, which attenuated insulin resistance, as well as liver inflammation and fibrosis, in preexisting steatohepatitis. These effects have been associated with a decrease in oxidative stress in cells.

### 5.4. Anti-Diabetic Effect of Lycopene

Lycopene appears to have beneficial effects in improving factors related to diabetes progression, including oxidative stress, inflammation, and endothelial dysfunction [226]. It was observed that the administration of lycopene in rats decreased glucose levels, increased insulin concentration, reduced H_2_O_2_, TBARS, and iNOS levels, increased cNOS activity and NO levels, as well as increased total antioxidant capacity with increased CAT, SOD, and GSH-Px activity [227,228,229].

In humans, in a placebo-controlled clinical trial with patients with type 2 diabetes mellitus, Neyestani et al. [230] found a negative correlation between total antioxidant capacity and MDA in the lycopene-treated group, indicating that lycopene supplementation attenuates oxidative stress in these patients. According to Yin et al. [231], lycopene strengthens the antioxidant defense system against oxidative stress, attenuating insulin signaling deficits, inhibiting neuroinflammation, and improving cognitive function. These studies suggest that lycopene may help improve the progression of diabetes in humans.

### 5.5. Anti-Cataract Effect of Lycopene

The ocular environment is rich in endogenous sources of RONS. Although there are several physiological defenses to protect ocular lenses from the toxic effects of light and oxidative damage, evidence suggests that long-term chronic exposure to oxidation can damage the lens and predispose it to the development of cataracts [232]. In this sense, Gupta et al. [233] showed that lycopene supplementation in rats restored GSH, SOD, CAT, and GST levels and, consequently, prevented sodium selenite-induced cataracts. According to the authors, lycopene protects against the experimental development of cataracts due to its antioxidant properties and may be useful for cataract prophylaxis or therapy [234]. Also, Göncü et al. [235] demonstrated the anti-inflammatory effect of lycopene on lipopolysaccharide-induced uveitis in rats. According to the authors, the anti-inflammatory activity of lycopene was mediated by the inhibition of TNF-α, NO, and IL-6 production, resulting in reduced inflammation and uveal oxidative stress.

### 5.6. Anti-Cancer Effects of Lycopene

Studies have shown that lycopene can reduce the risk of cancer by inducing antioxidant enzymes and phase II detoxifying enzymes such as NAD(P)H quinone oxidoreductase 1 and γ-glutamylcysteine synthetase [236]. These enzymes eliminate many harmful substances, converting them into hydrophilic metabolites that can be readily excreted from the body. In fact, lycopene administration significantly suppressed gastric cancer in vivo, reducing lipid peroxidation, increasing the levels of vitamin C, vitamin E, and GSH, and increasing circulating activity dependent on enzymes such as GSH-Px and GST [237]. Lycopene also prevented experimental oral carcinogenesis by inhibiting oxidative stress through the upregulation of detoxification pathways [238]. Recently, Cheng et al. [239] demonstrated the efficacy of lycopene in inhibiting the oxidative stress induced by cigarette smoke in lung cancer epithelial cells.

Other potentially beneficial effects of lycopene include inhibition of carcinogenic activation, proliferation, angiogenesis, invasion, and metastasis, blocking tumor cell cycle progression, and induction of apoptosis through its antioxidant activity and changes in various signaling pathways [240,241,242,243,244]. In addition, lycopene improved communication between cells by stimulating gap junctions, which is believed to be one of the protective mechanisms related to the cancer-preventive activities attributed to lycopene [195].

In in vitro studies, lycopene treatment selectively interfered with cell growth and induced apoptosis in cancer cells without affecting normal cells [200,245]. In vivo studies have shown the protective effects of lycopene against liver cell carcinoma and prostate cancer [197,237,246,247,248,249,250,251,252,253]. In addition to the correlation between lycopene and prostate cancer demonstrated in clinical studies, increasing evidence suggests that lycopene plays an important role in preventing cancer in other organs such as the breast, lung, gastrointestinal tract, pancreas, cervix, and ovaries [254,255,256,257].

## 6. Effects of Lycopene on Malaria

The use of antioxidant compounds in the treatment of tropical diseases has increased, including Chagas disease, dengue, and malaria, as several studies have suggested the involvement of oxidative stress in the pathogenesis and progression of these diseases [54,258,259]. In this context, studies show that the discovery of new antimalarial drugs is necessary, and natural antioxidant products are important sources for obtaining new antimalarial compounds or even as adjuvant therapy, enhancing the activity of antimalarial drugs [260,261,262,263,264].

A study by Metzger et al. [34] demonstrated that natural products can be used in malaria chemotherapy. According to this study, increased plasma lycopene concentration was associated with faster clearance of parasites in children. In a related study, Caulfield et al. [265] demonstrated that the nutritional deficiency of the host is associated with the morbidity and mortality of children with severe malaria. In this sense, previous studies suggest that changes in plasma concentrations of micronutrients, including vitamins A and C, retinol, β-carotene, α-carotene, β-cryptoxanthin, lutein, and lycopene, occur due to increased use of these antioxidants in patients with malaria, suggesting that there may be a need for vitamin supplementation in patients with malaria [266]. In corroborating this suggestion, the nutritional deficit seems to be associated with a redirection of these antioxidants to the liver to aid in the synthesis of acute-phase proteins in other organs, repair tissue damage caused by the infectious organism, and aid in the host’s oxidative defense mechanisms [33].

In fact, Sondo et al. [32] had already reported that periodic supplementation of high doses of vitamin A and zinc could reduce the morbidity caused by malaria. In this sense, Agarwal et al. [147] investigated the effect of lycopene on the growth of *P. falciparum* in vitro, monitoring the progression at different stages. These authors showed that lycopene treatment induced an increased production of RONS in the cytoplasm of the parasite, which caused the parasite to lose its mitochondrial membrane potential and cytotoxicity, resulting in merozoites not being released from the erythrocytes of the host, suggesting that the inclusion of lycopene in the diet may be useful in changing the clinical outcomes of malaria.

Preliminary results from our research group demonstrated that lycopene supplementation in mice (BALB/c; 3.11 mg/kg) infected with the *P. berghei* strain showed a delay in the induction and a decreased progression of parasitemia. Also, the animals supplemented with lycopene showed a higher rate of survival compared to the positive control [267], suggesting lycopene prophylactic and antiparasitic activity, which may be due to the cytotoxic effect of lycopene against the parasite [147], suggesting an important role of lycopene supplementation in preventing malaria [268].

### 6.1. Neuroprotective Effect of Lycopene

Individuals infected with *P. falciparum* can rapidly progress to severe anemia, respiratory distress, and cerebral malaria [269]. Cerebral malaria is associated with debilitating neurological impairments in survivors, as well as higher number of malaria deaths [270]. Although there is no complete understanding of the exact mechanisms and processes that lead to neuronal cell death in cerebral malaria, studies demonstrate that elevated levels of the inflammatory cytokines and RONS contribute to neuronal cell death in cerebral malaria [271,272].

Furthermore, considerable evidence suggests that microvascular dysfunction, sequestration of parasitized blood cells in the microcirculation, an abrupt reduction in blood flow, and cerebral hypoxia are essential for ischemic stroke, characterized by the presence of both ischemic and reperfusion-induced injuries in the brain, leading to neuronal dysfunction and death [273]. In this context, studies by Paul et al. [274] and Farouk et al. [275] point out that lycopene is a powerful antioxidant, permeable to the blood-brain barrier, with neuroprotective activity. Previously, Hsiao et al. [276] showed that treatment with lycopene in rats (4 mg/kg) prevented ischemic brain injury induced by middle cerebral artery occlusion by inhibiting microglia activation and NO production, resulting in reduced infarction volume in brain injury by the ischemia-reperfusion syndrome. Additionally, lycopene has been shown to protect the brain from ischemic damage by its ability to increase GSH production and decrease RONS production. Furthermore, lycopene activates the expression of nuclear factor erythroid 2 related factor 2 and heme oxygenase-1, one of the antioxidant pathways involved in the attenuation of oxidative stress and the maintenance of the redox state in various tissues and organs, such as the brain tissue [277].

Oxidative stress is also strongly implicated in the pathogenesis of neurodegenerative diseases, such as Alzheimer’s disease (AD) [278] and Parkinson’s disease (PD) [279]. In this sense, Kaur et al. [280] demonstrated that lycopene supplementation in rats (10 mg/kg) for 30 days was able to reduce oxidative stress in rotenone-induced PD, restoring GSH and SOD levels and reversing complex I inhibition of the electron transport chain, exerting a protective effect on motor and cognitive deficits. Furthermore, according to Prema et al. [281], lycopene induces increased expression of the antiapoptotic protein B-cell lymphoma 2 protein (BCL-2) and decreased release of the proapoptotic proteins cytochrome c, protein x associated with BCL-2 (BAX), and caspases-3, 8, and 9, preventing apoptosis in mice with PD induced by 1-methyl-4-phenyl-1,2,3,6-tetrahydropyridine.

Previously, studies have shown the activation of inositol-requiring enzyme 1, induction of X box-binding protein 1, upregulation of BAX, downregulation of BCL-2, and cleavage of caspase-3 indicating the endoplasmic reticulum stress-mediated apoptotic pathway in PbA-infected mouse brains involved in neuronal cell death in severe/cerebral malaria [271]. Thus, lycopene reverses neurochemical deficits, oxidative stress, apoptosis, and physiological abnormalities in malaria and PD-induced mice.

Other studies reinforce the importance of lycopene in neuronal mitochondrial function. In a rat cortical neuron culture model using an established paradigm of β-amyloid (Aβ) peptide-induced cell injury, Qu et al. [282] found that lycopene significantly inhibited intracellular RONS and prevented Aβ-induced mitochondrial fragmentation. Furthermore, it inhibited the opening of mitochondrial permeability transition pores as well as the release of cytochrome c. Lycopene also prevented a decrease in the enzymatic activity of the mitochondrial complex and a reduction in the generation of ATP, besides preventing the occurrence of damage to the mitochondrial DNA and improving the level of the mitochondrial transcription factor A in the mitochondria. These results suggest that the ability of lycopene to prevent Aβ-induced neurotoxicity is closely related to the inhibition of mitochondrial oxidative stress and improvement of mitochondrial function [282,283].

Behavioral experiments confirmed that lycopene consistently reduced Aβ accumulation in elderly CD-1 mice [284]. Lycopene also attenuated age-associated cognitive impairments, including those involving locomotor activity, working memory, and spatial cognitive memory. Lycopene administration reversed the systemic and oxidative stress responses of the central nervous system induced by aging. Furthermore, lycopene downregulated the expression of inflammatory mediators and prevented synaptic dysfunction in aged mouse brains [285]. Huang et al. [286] also showed the antagonistic effect of lycopene on neuronal oxidative damage induced by tert-butyl hydroperoxide in vitro. Moreover, lycopene increased cell viability, improved neuron morphology, increased GSH levels, and decreased the production of RONS. Lycopene also reduced the expression of BAX, cytochrome c, and caspase-3 and increased the expression of BCL-2 and phosphoinositide 3-kinase/Akt (PI3K/Akt) [286]. Recent studies confirm that lycopene prevents neuronal apoptosis through the activation of the PI3K/Akt signaling pathway, important regulators for preventing mitochondrial damage and apoptosis induced by oxidative stress, ischemia-reperfusion syndrome, that play an important role in severe/cerebral malaria [287,288,289].

### 6.2. Effects of Lycopene as an Immunomodulator

Other factors related to neuronal injury and death in severe/cerebral malaria include the release of RONS, mitochondrial dysfunction, induction of programmed cell death, microglia activation, and release of inflammatory mediators [290,291].

Studies indicate that in malaria infection, increased expression of high mobility group box-1 is observed, which interacts with cell surface receptors such as toll-like receptor-4 (TLR-4), leading to the overproduction of pro-inflammatory cytokines (IL-1β, IL-6, IL-12, TNF-α, and IFN-γ) and anti-inflammatory cytokines (IL-4, IL-10, and IL-13) [289,291,292,293]. The action of these cytokines in conjunction with disturbances present in the microcirculation can affect both the integrity and functions of the blood-brain barrier, leading to vascular congestion, disruption of the blood-brain barrier, cerebral edema, impaired perfusion, and neuronal damage [294,295].

Several studies have highlighted the ability of carotenoids and their metabolites to regulate intracellular signaling cascades, modulating gene expression and protein translation in metabolic pathways associated with inflammatory and oxidative stress [296,297]. In this sense, studies indicate that lycopene can modulate the production of IL-1β, TNF-α, IL-2, IL-10, and IFN-γ, exerting an immunomodulatory effect on the peripheral blood mononuclear cells of healthy individuals [298], as well as suppressing the production of NO, IL-6, and TNF-α [299,300]. According to Feng et al. [299] and Vasconcelos et al. [301], lycopene interferes with the phosphorylation of the inhibitory protein kappa B, protecting it from degradation and preventing the release and translocation of NF-κB, a transcription factor that plays an important role in regulating the expression of genes responsible for inflammation, such as TNF-α, IL-1β, iNOS, and COX-2, proliferation, and apoptosis (Figure 7).

Other studies reinforce that the blockade of NF-κB activation by lycopene appears not to be tissue- or cell-type specific and may represent a way in which lycopene can inhibit the production of other inflammatory mediators, including TNF-α, NO, and IL-6, resulting in reduced inflammation [302,303]. In this sense, Gouranton et al. [304] showed that lycopene reduced TNF-α-induced activation of the NF-κB signaling pathway in adipocytes. According to the authors, this effect was fundamental for the TNF-α-mediated decrease in the expression of proinflammatory cytokines and chemokines in adipocytes and pre-adipocyte 3T3-L1 cells. The same effect was observed in human adipocytes, where lycopene decreased the expression of IL-6, monocyte chemotactic protein 1, and IL-1β induced by TNF-α [304].

The reduced production of the proinflammatory cytokines IL-1β and TNF-α, as well as the increased secretion of the anti-inflammatory cytokine IL-10, indicate that lycopene can boost anti-inflammatory responses [305]. In addition, lycopene can increase IL-12 and IFN-γ secretion in human peripheral blood mononuclear cells, indicating that lycopene enhances the immune response of the host [306]. In this sense, evidence from an ex vivo study indicates the stimulatory effect of lycopene on cytokine production by T-helper 1 lymphocytes, resulting in a cell-mediated immune response. Yamaguchi et al. [307] observed that the oral administration of lycopene in mice (5 mg/kg/day) significantly suppressed capsaicin-induced production of IL-2, IFN-γ, and IL-4 in lymphoid tissue cells in the small intestine wall, cytokines that are involved in the development of immunity to the antigens present there. Furthermore, lycopene did not alter the T lymphocyte population, indicating that lycopene accelerates and/or suppresses T-helper cytokines in these cells, acting to modulate the immune response.

Other studies have also verified the potential anti-inflammatory effect of lycopene combined with other substances, such as lutein, omega-3, and carnosic acid [308,309]. In this sense, Phan et al. [310] showed a reduction in IL-8 secretion by human colorectal Caco-2 cells in the presence of lycopene and anthocyanin mixtures. Previous studies have shown that the association with substances such as lutein, selenium, and β-carotene promotes a synergistic effect, intensifying NO, TNF-α, SOD, and prostaglandin E2 production inhibition, as well as MDA derived from the down-regulation of iNOS, COX-2, NADPH oxidase, or 5-lipoxygenase expression, and inhibition of TNF-α secretion [311,312].

Together, these data support the anti-inflammatory and immunomodulatory effect of lycopene on major cell subtypes, namely, adipocytes, pre-adipocytes, and macrophages, cells that are involved in the production of inflammatory cytokines and chemokines in malaria.

## 7. Future Trends and Conclusions

The benefits provided by lycopene can be attributed mainly to its direct antioxidant activity [229]. This activity is generally responsible for protecting the cellular system from a variety of RONS, including ^1^O_2_, O_2_^•−^, NO, and OONO^-^, as well as having an indirect action through the upregulation of antioxidant substances, in addition to preventing other diseases [313]. Finally, Figure 8 summarizes the mechanisms of action considered in the present review.

Furthermore, the antioxidant status of lycopene, similar to other carotenoids, has also been implicated in the pathogenesis of malaria in vitro and in vivo [33]. In a previous study, Agarwal et al. [147] showed the cytotoxic effects of lycopene against *P. falciparum* in vitro, suggesting an important role of lycopene in preventing malaria [268]. Although treatment regimens with various antimalarials are used in clinical practice, there are still no substances that can prevent the disease. Thus, it can be suggested that dietary lycopene may be useful in changing the clinical outcomes of malaria. This review provides evidence of the antioxidant and anti-inflammatory benefits of lycopene supplementation, therefore suggesting it be included when formulating new prevention strategies to fight malaria and several other diseases.

## Figures and Tables

**Figure 1 nutrients-14-05303-f001:**
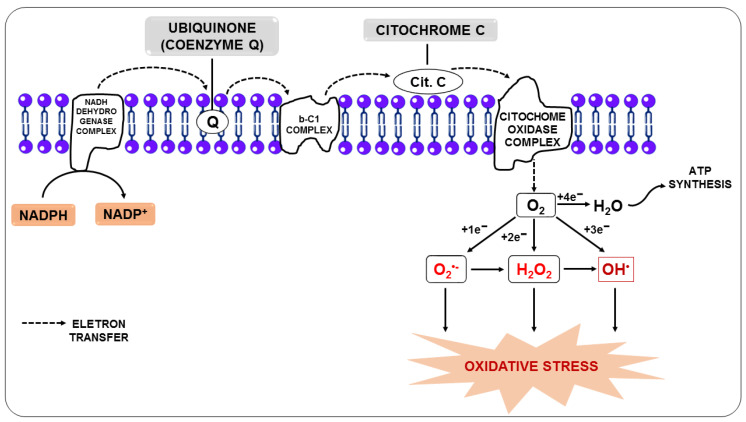
Production of reactive oxygen species from the transfer of electrons from the electron transport chain.

**Figure 2 nutrients-14-05303-f002:**
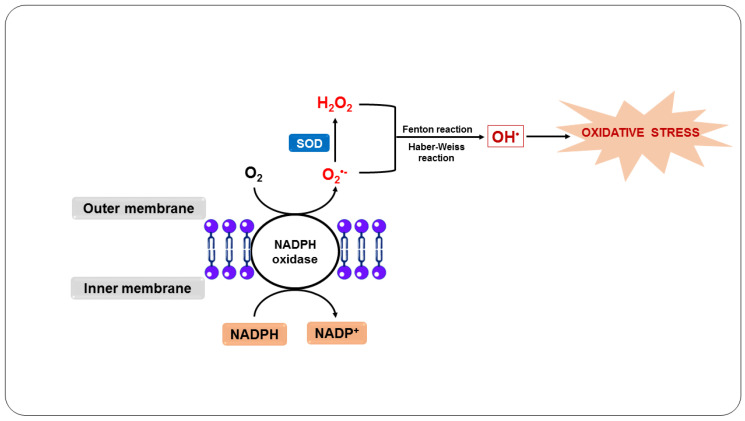
Oxidative stress as a host defense mechanism in response to infection by *Plasmodium* sp.

**Figure 3 nutrients-14-05303-f003:**
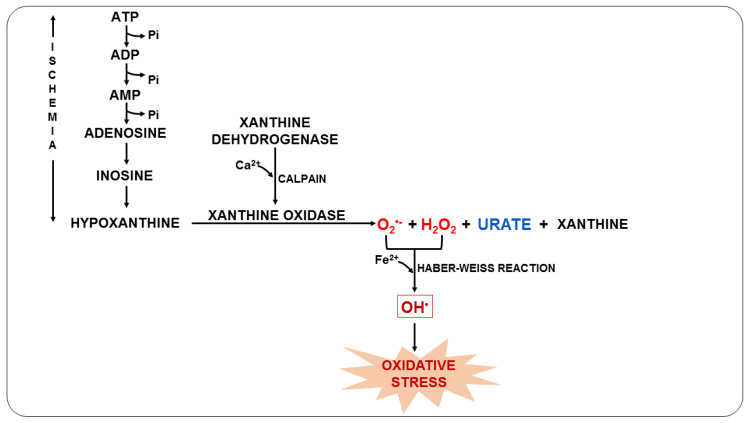
Oxidative stress due to ischemia-reperfusion syndrome during malaria.

**Figure 4 nutrients-14-05303-f004:**
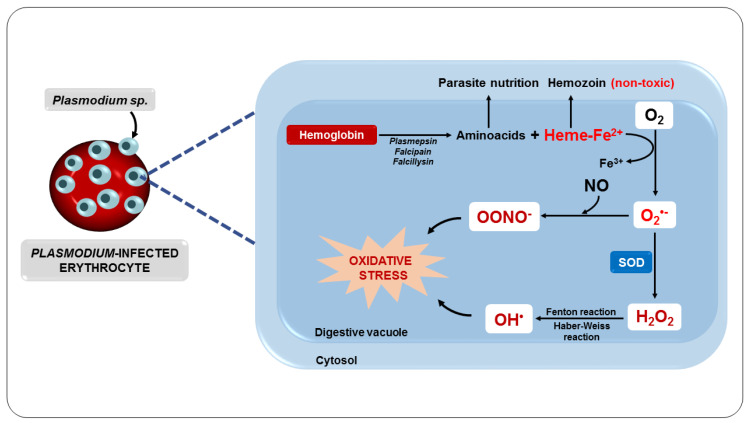
Oxidative stress as a consequence of parasite metabolism.

**Figure 5 nutrients-14-05303-f005:**
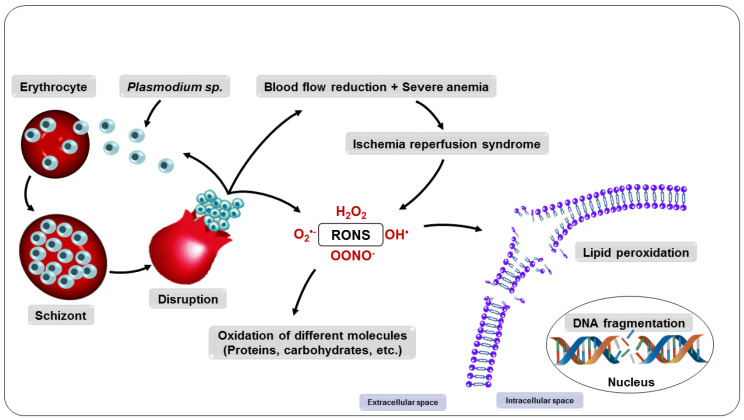
Consequences of the multiplication of parasites in the erythrocyte.

**Figure 6 nutrients-14-05303-f006:**
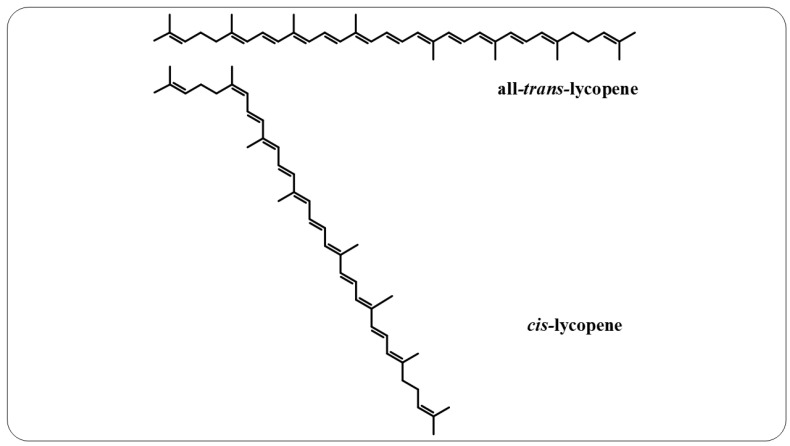
All-*trans*-lycopene and *cis*-lycopene structures.

**Figure 7 nutrients-14-05303-f007:**
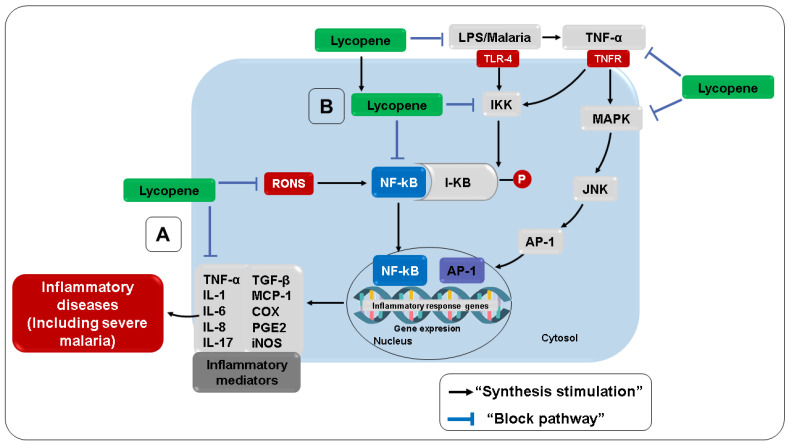
Anti-inflammatory effects of lycopene. (A) Direct anti-inflammatory activity. (B) Indirect anti-inflammatory activity. TGF-β, Transforming growth factor-beta; AP-1, activator protein-1; JNK, c-jun *N*-terminal kinase; MAPK, mitogen-activated protein kinases; I-κB, kappa B inhibitory protein; LPS, lipopolysaccharide; TNF-α, tumor necrosis factor-alpha; IL-1, interleukin 1; IL-6, interleukin 6; MCP-1, monocyte chemoattractant protein 1.

**Figure 8 nutrients-14-05303-f008:**
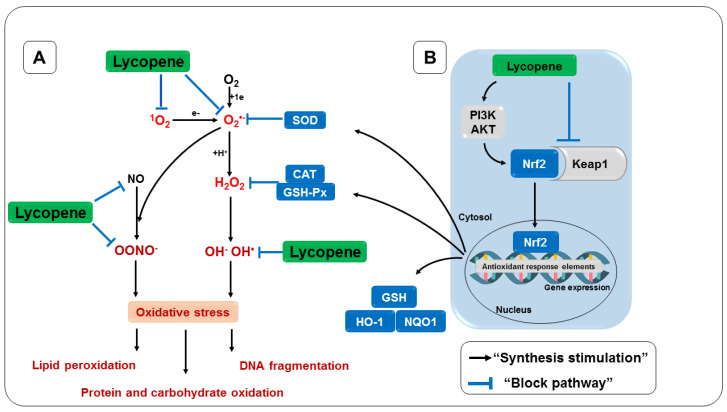
Antioxidant effect of lycopene. (**A**) Direct antioxidant activity. (**B**) Indirect antioxidant activity. Keap1, Kelch-like inhibitory protein 1; Nrf2, erythroid nuclear factor 2; GSH, glutathione; SOD, superoxide dismutase; CAT, catalase; GSH-Px, glutathione peroxidase; PI3K/AKT, phosphoinositide 3-kinase/AKT.

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
