# Peer review of "Potential Benefits of Lycopene Consumption: Rationale for Using It as an Adjuvant Treatment for Malaria Patients and in Several Diseases"

_nutrients, 2022, doi:10.3390/nu14245303_

Round 1
Reviewer 1 Report
Comments and Suggestions for Authors
Dear Editor,
I have reveiwed the manuscript entiteled Potential benefits of lycopene consumption: rationale for using it as an adjuvant treatment for malaria patients and in several diseases submitted for publication in Nutrients. I find the manuscript interesting and of moderate imporantace for the audence, since lycopene is quite well studied and reviewed in the past, however, I think that the manuscript could be suitable for publication after some revision. Bellow are the comments which should be followed during the process of manuscript reviewing and editing:
Only one reference from 2022, please find some contemporary referneces and do remove those from early 2000s.
One paragraph should consist out of 5 sentences at least, please follow this rule.
Please reduce the lenght of the text which is not in direct relation to antimalarial activity of lycopen, or some activity asociated with complications of malaria. This text is too long and you are losing the point of antimalaria activity of lycopen as the title of review suggests. If you prefere you can change the title of the paper since at the moment it does not corelate with the text.
I would be happy if you can provide sturctures of antimalarial drugs and their metabolites causing oxidative damage or any kind of damage as such, with highlighted part(s) of molecule asociated with the obeserved acitivity (if applicable).
Reviewer 2 Report
Comments and Suggestions for Authors
accept
Author Response
Dear reviewer,
On behalf of all the authors, we thank you and the reviewers for the acceptance of our manuscript and for the comprehensive review, which resulted in considerable improvement of the manuscript.
Sincerely,
Everton Varela, PhD and Sandro Percario, DSc, PhD
Reviewer 3 Report
Comments and Suggestions for Authors
This review mainly discussed the potential of lycopene, a kind of carotenoids, to suppress inflammation caused by malaria infection from the perspective of antioxidant mechanisms.
The reviewer mainly comments on the bioavailability of lycopene, and also adds one minor comment.
1, “4. Lycopene” section
The structural formula of lycopene is shown in Figure 6. The cis isomer looks like the cis 6 (or 6') isomer of lycopene. However, this cis isomer is very minor among lycopene cis isomers. The main ones are the cis 5, 9, 13, and 11 positions. Why did the authors draw the structural formula of the very minor cis 6 position? Does only this cis isomer have a special effect?
2, “4.2. Absorption” subsection
“During absorption, lycopene taken up by the enterocyte can also be cleaved by beta-carotene 9’,10’-oxygenase (BCO2) to form apo-10’-carotenoid metabolites, including lycopene apo-10’-lycopenols [181–183].”
“In addition, several apolycopenols have been isolated from fruits, vegetables, and human plasma [191].”
References 181-182 are documents from animal experiments, and 183 is a document from a human trial, but they do not consider apolycopenol.
In reference 191, apolycopenal (not apolycopenol) is detected in human blood, but it originates from ingested substances and may be not a metabolic product in humans.
After reading these sentences, the reader might think that lycopene is also metabolized to apolycopenal (or even apolycopenol, apolycopenoic acid) after absorption in humans.
Are there any recent findings on whether the expression of BCO2 in humans or the products of this enzyme can be detected in the blood or elsewhere? If so, please add them.
3, “5.3. Hepatoprotective Effect of Lycopene” subsection
“Similar results were observed in the study by Abdel-Daim et al. [225], where zinc oxide poisoning in fish caused severe lipid peroxidation with a significant increase in the level of MDA in the liver, kidney, and gill tissues, and treatment with lycopene significantly reduced the production of this oxidative stress biomarker”
There is a period missing from the end of this sentence.
4, “6. Effects of Lycopene on Malaria” section
“In this sense, previous studies suggest that changes in plasma concentrations of micronutrients, including vitamins A and C, retinol, a-carotene, b-carotene, b-cryptoxanthin, lutein, and lycopene, occur due to increased use of these antioxidants in patients with malaria, suggesting that there may be a need for vitamin supplementation in patients with malaria [266].”
Although it says "vitamins A and C, retinol," vitamin A and retinol are the same. There is a duplication. There is no need to mention either one.
5, Reference 33
There are few references on human studies of lycopene and malaria (only reference 33?). Reference 33 appears to be an epidemiological study on the effects of retinol and various carotenoids (α-carotene, β-carotene, β-cryptoxanthin, lutein, and lycopene) against malaria. Are there any other reports of research (epidemiological studies or intervention trials) on the effects of lycopene against malaria in humans? Please cite if any.
6, “6.1. Neuroprotective Effect of Lycopene” subsection
“In this context, studies by Paul et al. [274] and Farouk et al. [275] point out that lycopene is a powerful antioxidant, permeable to the blood-brain barrier, with neuroprotective activity.”
Reference 274 is a review article. Reference 275 is an experiment using rats.
What readers may be interested in is whether lycopene can cross the blood-brain barrier in humans. Is there any report of this? Please cite if any.
7, “7. Future Trends and Conclusions” section
“suggesting an important role of lycopene in preventing malaria [268].”
Reference 268 is an intervention study of vitamin A and zinc, so lycopene seems irrelevant. There is a mismatch between the content of the main text and the cited references.
